# Cold Ischemia Time and Graft Fibrosis Are Associated with Autoantibodies after Pediatric Liver Transplantation: A Retrospective Cohort Study of the European Reference Network *TransplantChild*

**DOI:** 10.3390/children9020275

**Published:** 2022-02-17

**Authors:** Norman Junge, Angelo Di Giorgio, Muriel Girard, Zeynep Demir, Diana Kaminska, Maria Janowska, Vaidotas Urbonas, Dominykas Varnas, Giuseppe Maggiore, Tommaso Alterio, Christoph Leiskau, Florian W. R. Vondran, Nicolas Richter, Lorenzo D’Antiga, Rafael Mikolajczyk, Eva-Doreen Pfister, Ulrich Baumann

**Affiliations:** 1Division for Pediatric Gastroenterology and Hepatology, Department of Peadiatric Kidney, Liver, and Metabolic Diseases, Hannover Medical School, 30625 Hannover, Germany; leiskau.christoph@mh-hannover.de (C.L.); pfister.eva-doreen@mh-hannover.de (E.-D.P.); baumann.u@mh-hannover.de (U.B.); 2Department of Paediatric Hepatology, Gastroenterology and Transplantation, Azienda Ospedaliera Papa Giovanni XXIII, 24127 Bergamo, Italy; adigiorgio@asst-pg23.it (A.D.G.); ldantiga@asst-pg23.it (L.D.); 3Hépatologie Pédiatrique–Transplantation Hépatique, Hospital Necker Enfants-Malades, 75015 Paris, France; muriel.girard@aphp.fr (M.G.); zeynep.demir-ext@aphp.fr (Z.D.); 4Department of Gastroenterology, Hepatology, Nutritional Disorders and Pediatrics, The Children’s Memorial Health Institute, 04-730 Warsaw, Poland; diana.kam@wp.pl; 5Department of Pediatric Surgery & Organ Transplantation, The Children’s Memorial Health Institute, 04-730 Warsaw, Poland; m.janowska@ipczd.pl; 6Clinic of Children’s Diseases, Institute of Clinical Medicine, Faculty of Medicine, Vilnius University, LT-03101 Vilnius, Lithuania; vaidotas.urbonas@santa.lt (V.U.); dominykas.varnas@santa.lt (D.V.); 7Gastrointestinal, Liver, Nutrition Disorders Unit, Liver Transplantation Center, IRCCS Pediatric Hospital Bambino Gesù, 00165 Rome, Italy; giuseppe.maggiore@opbg.net (G.M.); tommaso.alterio@opbg.net (T.A.); 8Department of Pediatrics and Adolescent Medicine, Division of Pediatric Neurology, University Medical Center Göttingen, Georg August University, 37075 Göttingen, Germany; 9Department of General, Visceral and Transplant Surgery, Hannover Medical School, 30625 Hannover, Germany; vondran.florian@mh-hannover.de (F.W.R.V.); richter.nicolas@mh-hannover.de (N.R.); 10Institute for Medical Epidemiology, Biometrics and Informatics, Interdisciplinary Center for Health Sciences, Medical Faculty of the Martin Luther University Halle-Wittenberg, 06112 Halle, Germany; rafael.mikolajczyk@uk-halle.de

**Keywords:** T-cell mediated rejection, liver biopsy, de novo autoimmune hepatitis, graft fibrosis, chronic rejection, antinuclear antibody, smooth muscle antibody, liver-kidney-microsome antibody, pediatric liver transplantation

## Abstract

The reported prevalence of autoantibodies (AAB) (ANA, SMA, LKM, SLA) after pediatric liver transplantation (pLTX) varies considerably from 26–75%, but their clinical impact on outcome is uncertain. We aimed to study the prevalence of AAB after pLTX, their association with donor-, transplant-, and recipient-characteristics, and their relation to outcome. In our multicenter retrospective study, we aimed to clarify conflicting results from earlier studies. Six ERN *TransplantChild* centers reported data on 242 patients, of whom 61% were AAB positive. Prevalence varied across these centers. Independent of the interval between pLTX and AAB analysis, a one-hour increase in CIT resulted in an odds ratio (OR) of 1.37 (95% CI 1.11–1.69) for SMA positivity and an OR of 1.42 (95%CI 1.18–1.72) for ANA positivity. Steroid-free immunosuppression (IS) versus steroid-including IS (OR 5.28; 95% CI 1.45–19.28) was a risk factor for SMA positivity. Liver enzymes were not associated with ANA or SMA positivity. We did not observe an association of rejection activity index with ANA or SMA. However, the liver fibrosis score in follow-up biopsies was associated with ANA titer and donor age. In conclusion, this first multicenter study on AAB after pLTX showed high AAB prevalence and varied widely between centers. Longer CIT and prednisolone-free-IS were associated with AAB positivity, whereas AAB were not indicative of rejection, but instead were associated with graft fibrosis. The detection of AAB may be a marker of liver fibrosis and may be taken into consideration when indications for liver biopsy and immunosuppressive regimes, or reduction of immunosuppression in long-term follow-up, are being discussed. Prospective immunological profiling of pLTX patients, including AAB, is important to further improve our understanding of transplant immunology and silent graft fibrosis.

## 1. Introduction

Pediatric liver transplantation (pLTX) is an excellent treatment option for many acute or chronic and congenital or acquired liver diseases, with a five-year patient and graft survival rate of 97.5% and 86.5%, respectively [1]. However, the long-term outcome is determined by the side effects of immunosuppression or silent graft fibrosis. This implies that individually balanced immunosuppression is of major importance. Whilst there are no reliable non-invasive markers for this, autoantibodies (AAB) associated with autoimmune liver disease (AILD), such as antinuclear antibody (ANA), smooth muscle antibody (SMA), liver-kidney-microsome antibody (LKM), and soluble liver antigen (SLA), are described as associated with graft fibrosis after pLTX [2,3]. AAB, in combination with other diagnostic parameters from the autoimmune hepatitis score [4], are good markers of AILD, but they are not specific and can be present in many acute [5,6] and chronic liver diseases [7,8,9]. Findings on the prevalence of AAB after pLTX in previous studies are inconsistent, with the variation of AAB prevalence from 26–75% [10,11,12,13]. Similarly, results on the clinical impact of AAB differ among studies. Some studies found an association between AAB and chronic rejection [11,13] or chronic graft hepatitis and fibrosis [2,3], while others did not [14,15]. Some studies connected their presence with different patient, donor, or transplant procedure characteristics [16] or biochemical graft injury [10,16]. Furthermore, studies on patients with non-alcoholic fatty liver disease [8] or alcoholic liver disease [9] showed an association of AAB with advanced liver disease. Due to inconsistent results in this field and a lack of multicenter studies, we felt that this topic deserved further exploration. We aimed to expand current knowledge on the prevalence, associated factors, and clinical impact of AAB after pLTX in a European multicenter study.

## 2. Materials and Methods

This analysis is based on a retrospective multicenter cohort study, including patients from 6 European transplant centers (Bergamo, Hannover, Paris, Rome, Vilnius, and Warsaw), within the European Reference Network (ERN) *TransplantChild*. Liver-transplanted patients with available AAB analyses (AABa) at any time during follow-up after pLTX, but at least 3 months after, were included. In patients with multiple pLTX, the most recent pLTX before AABa was used to define transplant related data and donor characteristics, so that the follow-up data of the most recent pLTX was not mixed with the transplant and donor characteristics of an earlier transplant. Patients who died within the first three months, or only had AABa within the first three months after pLTX, and patients with ANA or SMA analysis by ELISA only were excluded. Additionally, patients with combined transplantations were excluded. Medical data were extracted from medical records by local ERN investigators. Serum liver enzymes were collected in U/L.

One AAB measurement at any time after pLTX (by indication or by protocol) was included for every patient. In some of these patients, the AAB measurements were combined with liver biopsy (by indication or by protocol). AAB testing was performed at each center according to standard methods recommended by current international guidelines [17]. ANA, SMA, and anti-LKM analyses were performed by indirect immunofluorescence on rodent liver, stomach, and kidney sections. ANA positivity was defined as a titer ≥ 1:40 and SMA positivity as a titer ≥ 1:20. SLA was established by ELISA.

Liver histology was evaluated by local, experienced liver pathologists from each participating transplant center. Where available, T-cell mediated rejection (TCMR) was staged by rejection activity index (RAI) [18] and fibrosis by ISHAK Fibrosis Score [19]. Detailed information on RAI and ISHAK Fibrosis score is shown in the Appendix A. Liver biopsy data for RAI was collected if liver biopsy and AAB measurement were carried out within a timeframe of 30 days. ISHAK Fibrosis score data was collected on the patients’ latest liver biopsy on follow-up, where available.

For statistical analysis, IBM SPSS 26 was used. For multivariable analysis of the factors associated with ANA positivity (titer ≥ 1:40) and SMA positivity (titer ≥ 1:20), we applied binary logistic regression with the following variables: patient sex and age at pLTX, donor sex and age, CIT in hours, interval between pLTX and AABa in months, split liver versus whole organ, living donation liver transplantation (LDLTX) versus deceased donor (DDLTX), prednisolone-free immunosuppression versus immunosuppression with added prednisolone, aspartate aminotransferase (AST in U/L), alanine aminotransferase (ALT in U/L), and gamma glutamyl transferase (GGT in U/L). Linearity was assessed using the BoxTidwell procedure. Presence of multi collinearity was assesed based on correlation matrix, tolerance, and variance inflation factor. To test the correlation of levels of ANA and SMA titers with duration of CIT, we used the Spearman Rank correlation. This analysis was done in a subgroup of patients with an pLTX-AABa interval of 24–48 months.

For multivariable analysis of the factors associated with higher RAI in liver biopsies, linear regression with the following variables was applied: patients’ sex and age at pLTX, interval pLTX-AABa, living donor LTX (LDLTX), ANA titer, and SMA titer. In a second analysis, ANA and SMA titers were replaced by the dichotomous variables ANA > 1:80 and SMA > 1:80. Donor data and CIT, which were not documented for all patients in this study, were not included in this analysis to avoid a substantial reduction of analyzed patients.

For multivariable analysis of the factors associated with higher fibrosis scores in follow-up liver biopsies, linear regression with the following variables was applied: patients’ sex and age at pLTX, donor sex and age, CIT in hours, pLTX-AABa interval, ANA titer, and SMA titer. In a second analysis, ANA and SMA titers were replaced by the dichotomous variables ANA > 1:80 and SMA > 1:80. This approach was applied to test whether higher ANA and SMA titers have a stronger impact.

If not otherwise mentioned, given numbers for the statistical results always follow the sequence median, mean, and 95% confidence interval. As not all data for patients were available or applicable, the number of patients included is shown for each analysis. This study was performed in accordance with the Declaration of Helsinki. This observational study was performed completely anonymous and retrospectively, therefore ethical approval was not necessary. This report is in accordance with STROBE Guidelines [20].

## 3. Results

### 3.1. Cohort Characteristic

We included 242 patients from 6 pLTX centers in Europe participating in ERN *TransplantChild*. The detailed baseline characteristics are given in Table 1. As we could detect no patients with SLA and only one with LKM, we focused our analyses on ANA and SMA.

We observed differences between centers, regarding AAB prevalence and mean titers, as well as pLTX-AABa interval and transplant related data (CIT, graft and donation type). Standard immunosuppression in all centers was calcineurin-inhibition based, and alternative immunosuppression was in n = 11 children mTOR inhibition, n = 1 MMF, and n = 1 prednisolone only. Further immunosuppressive co-medication varied considerably between centers. The severity and frequency of graft fibrosis in the long-term follow-up biopsies differed only slightly between centers (Appendix A).

### 3.2. Variables Associated with SMA and ANA Positivitiy

In the multivariable model, developing titers >1:20 after pLTX for SMA was associated with longer CIT, living donor pLTX, younger donor age, and being treated with Prednisolon (Table 2). A one-hour increase of CIT increased the risk of having SMA >1:20 after pLTX with an odds ratio (OR) of 1.36 (95%CI: 1.11–1.69). The OR for SMA positivity in patients with prednisolone-free immunosuppressive treatment was 5.28 (95%CI: 1.45–19.28) and 37.05 (95%CI: 5.56–246.77) for patients who received a living donor LTX (LDLTX). Donor and patient sex, pLTX-AABa interval, and liver enzymes were not associated with SMA positivity.

Similarly, developing titer > 1:40 after pLTX for ANA was associated with longer CIT and living donor pLTX (Table 3). Donor or patient sex, pLTX-AABa interval, and liver enzymes were not associated with ANA positivity.

The above results are based on ANA and SMA positivity. To test whether titer levels correlate with duration of CIT, we tested for correlation of ANA and SMA titer levels with CIT in a sub cohort of patients whose pLTX-AABa interval was 24–48 months (to reduce the risk of the confusing effect of the interval). We found a correlation of CIT with ANA titers (Spearman’s rho = 0.464, *p* = 0.002) and SMA titers (rho = 0.346, *p* = 0.025).

### 3.3. Liver Disease Leading to pLTX

In our patients, the type of underlying disease leading to pLTX showed only minor differences in terms of ANA and SMA positivity (Figure 1).

### 3.4. Association between Autoantibodies and Histology Results

#### 3.4.1. T-cell Mediated Rejection

We analyzed 112 histology results from liver biopsies performed within 30 days of AABa. Multivariable analysis could not identify any variable associated with levels of RAI. In particular, ANA and SMA titers (Table 4), as well as ANA and SMA > 1:80 (Table 5), were not associated with levels of RAI in liver biopsy.

#### 3.4.2. Graft Fibrosis

We identified 78 patients with an available ISHAK fibrosis score from follow- up liver biopsies. The mean time from pLTX to liver biopsy was 103.1 months (median 87.0, 95%CI: 91.6–114.7). In a multivariable analysis, ANA titer, ANA > 1:80 versus ≤ 1:80, donor age, and CIT were associated with the level of ISHAK fibrosis score (Table 6 and Table 7).

## 4. Discussion

This is the first retrospective, multicenter study on AAB in liver-transplanted children. To date, existing studies report conflicting results. Our data confirms that AAB positivity is a frequent finding after pLTX (SMA positivity in 52.0% and ANA positivity in 26.7% of patients). Higher titers of ≥1:160 are more frequent for ANA than for SMA. The presence and titers of AAB were associated with CIT. The RAI of acute T-cell mediated reactions (TCMR) were not associated with AAB, but instead with a long-term follow-up graft fibrosis.

Our data on ANA and SMA positivity confirms previous studies (Richter et al. [12], Chen et al. [10]), but reports a higher prevalence than that published by Feng et al. [14] (26% ANA positive, 4% SMA positive), Saelans et al. [16] (43.3% AAB positive, 24% ANA positive, 21% SMA positive), and Avitzur et al. [11] (26% AAB positive).

Our study has made a number of important additional findings: one of the most important being the association between CIT, ANA, and SMA positivity. Furthermore, longer CIT was correlated with higher ANA and SMA titers. The association of AAB positivity and CIT could explain differences in AAB prevalence among earlier single center studies, as CIT is dependent on local requirements, regulations, and transplant procedures, such as frequency of living-donation and the distances deceased donor grafts have to be transported. Longer CIT is described as a risk factor for worse outcome after LTX in many studies [21,22] reviewed by Stahl JE et al. [23]. This poorer outcome could be explained by injury to hepatic sinusoidal epithelial cells leading to impaired microcirculation, ischemia-reperfusion damage, and release of inflammatory cytokines [23]. This ends in tissue inflammation and necrosis [21], which in turn could lead to the presentation of hepatic epitopes in concert with an inflammatory environment, which could initiate AAB generation and be the reason for the high prevalence of AAB. However, our retrospective study design is insufficient to show or prove any causal mechanisms. Nevertheless, the association between graft fibrosis and CIT, and ANA and SMA positivity is striking. In our study, the ISHAK fibrosis score in the multiple linear regression model was associated with ANA titers and CIT, which is in line with the results of Rhu et al. [24]. Sheikh et al. [15] could not confirm these results in a smaller group of patients, but AAB prevalence was lower in their cohort compared to ours.

It is important to note that even though an increase in AAB prevalence with increasing time after pLTX was described in earlier studies [11], in our analysis the relationship between AAB positivity and CIT persisted even after adjustment for the pLTX-AAB interval.

In addition to CIT, we could identify prednisolone-free IS as associated with SMA and ANA positivity. That prednisolone-free-IS was associated with AAB presence could suggest that IS treatment should be tailored individually and ideally based on liver histology results. Furthermore, LDLTX and younger donor age was associated with SMA positivity. Apart from these, no donor or patient variables, or liver enzyme levels, were associated with ANA or SMA positivity.

Underlying diseases, which were the indications for pLTX, showed no association to AAB after pLTX in our cohort. This is in contrast with the results of Kappi et al. [25]. They described a lower prevalence of AAB in patients with biliary atresia. However, others [10,11] reported no association of the underlying disease with pLTX and AAB after pLTX. Surprisingly, in our cohort, patients with autoimmune liver disease showed no substantially increased AAB positivity after pLTX, compared to other patients. This could be explained by the IS medication, as these patients are more likely to be treated with two IS medications, starting from pLTX.

We could not find an association of AAB and AST, ALT or GGT, as described by others [3,11,12,16]. This, together with the association of AAB with CIT and graft fibrosis, underlines the hypothesis that AAB, especially SMA, could be a marker for silent graft fibrosis.

Our results agree with those of Evans et al. [2]. They showed that AAB positivity was associated with chronic hepatitis leading to fibrosis, while AST activity was not associated with chronic hepatitis. Of course, these associations cannot be proven causatively, but, even if AAB only reflect the otherwise silent chronic graft hepatitis, they are of clinical relevance in the long-term care of pLTX patients and should be evaluated routinely, i.e., once yearly in follow-up. Benefits of steroid-free or reduced immunosuppression have to be balanced against the risk of long-term graft fibrosis on an individual basis and for that purpose AAB could be a part of the puzzle. The association of CIT with AAB positivity and fibrosis implies that there should be ongoing effort to minimize CIT and/or to minimize the impact of CIT on liver grafts. The latter is a fast-growing field of research with studies showing the benefit of hypothermic oxygenated perfusion (HOPE) prior to adult liver transplantation [26]. For pLTX, only case reports exist [27,28]. Further studies should look to AAB as a potential marker of benefit.

### Limitations

The main limitation of our study is also its strength: the multicenter design of the study which creates an inhomogeneous sample of patients. On the one hand, center differences, circumstances, and treatment protocols can be a confounder in the analysis, on the other hand, the findings apply to less selected patients. Another limitation is the retrospective design of our study and that, at the time, AAB analysis was not standardized and the interval between AAB analysis and pLTX, or long-term follow-up liver biopsy, varied. Furthermore, histological results can have an inter-observer variation. We cannot exclude pre-existing AAB before pLTX or from first pLTX in patients with multiple pLTX. An ideal prospective study would also analyze AAB before (included) pLTX, which is a major undertaking. However, our data could be a base to develop such a prospective study.

## 5. Conclusions

In conclusion, this first multicenter study on AAB after pLTX shows high AAB prevalence. CIT and prednisolone-free-IS were associated with higher AAB. AAB were not indicative of TCMR and were not associated with elevated AST, ALT, or GGT, but instead were associated with graft fibrosis, a finding which suggests that AAB can be potential, inexpensive, and non-invasive markers for silent graft fibrosis. Therefore, they should be measured on a regular basis at follow-up and should be taken into consideration when indications for liver biopsy and immunosuppressive regimes, or reduction of immunosuppression in long-term follow-up, are being discussed, but further research is needed. Prospective immunological profiling of pLTX patients, including AAB, is important to further improve our understanding of transplant immunology and silent graft fibrosis. Studies with follow-up liver biopsies should simultaneously include AAB analysis.

## Figures and Tables

**Figure 1 children-09-00275-f001:**
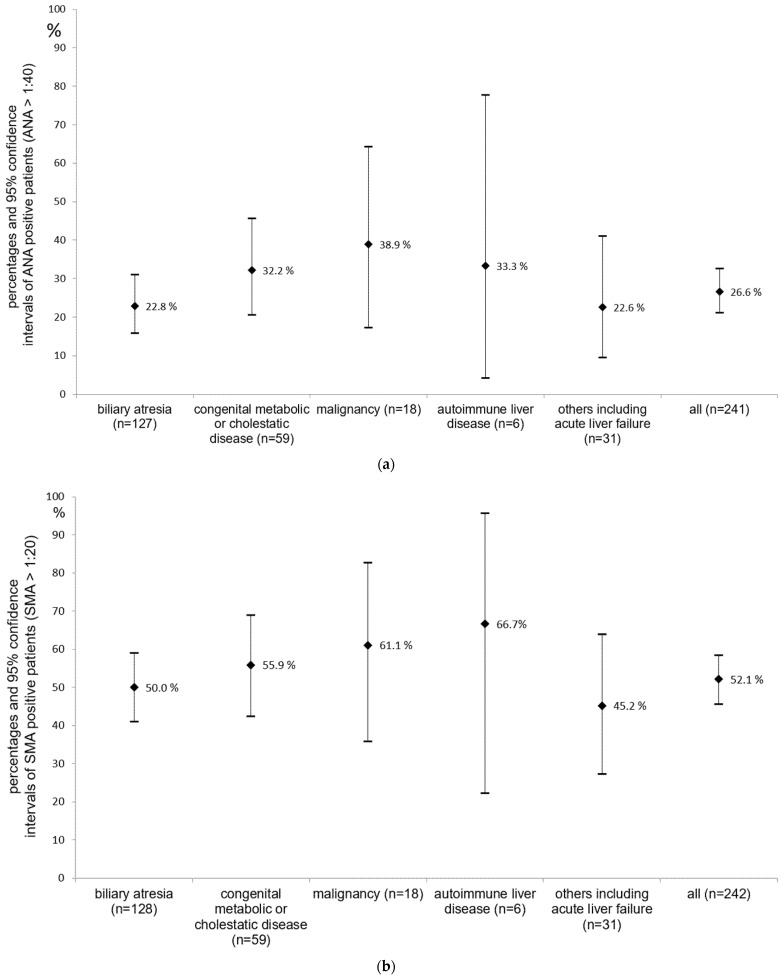
(**a**) ANA positivity depending on disease (percentages and 95% confidence intervals). (**b**) SMA positivity according to disease (percentages and 95% confidence intervals).

**Table 1 children-09-00275-t001:** Baseline characteristics of total cohort (n = 242).

Variable (n = Available Patients/Data)	Median (Mean); or Characteristic and Corresponding Absolute Number	95%Ci; or Percentage of Available Patients/Data	Minimal– Maximal Value
Sex (n = 242)	Female n = 132	54.5%	-
Age at pLTX (n = 242)	1.0 (3.1) y	2.6–3.7 y	0.0–17.3 y
Age at AAB analysis (n = 242)	7.0 (8.0) y	7.4–8.7 y	0.8–22.2 y
pLTX-AAB interval (n = 242)	49.0 (57.8) mo	52.2–63.3 mo	4.0–197.0 mo
Diagnosis (n = 242)	Biliary atresia n = 128	52.9%	-
	Genetic (Metabolic + Cholestatic) liver disease n = 59	24.4%	-
	Malignancy n = 18	7.4%	-
	Acute liver failure n = 9	3.7%	-
	Autoimmune liver disease n = 6	2.5%	-
	Other diagnoses n = 22	9.1%	-
Type of donation (n = 242)	Living-donor donation n = 44	18.2%	-
	Deceased-donor donation n = 198	81.8%	-
Type of Graft (n = 242)	Split SII + III° n = 185	76.5%	-
	Other split/reduced size n = 10	4.1%	-
	Whole liver n = 47	19.4%	-
Donor sex (n = 148)	Female n = 70	47.3%	-
Donor/Recipient sex match (n = 148)	Same sex n = 73	49.3%	-
Donor age (n = 149)	26.0 (26.0) y	23.6–28.4 y	0.0–59.0 y
Donor/Recipient age difference (n = 149)	24.0 (23.3) y	20.7–25.9 y	−4.0–53.0 y
Cold ischemia time (CIT) (n = 149)	387.0 (433.3) min	403.0–463.6 min	145.0–1025.0 min
CIT living donor pLTX (n = 28)	219.0 (220.6) min	203.2–238.0 min	145.0–300.0 min
CIT deceased donor pLTX (n = 125)	470.0 (488.0) min	457.4–518.5 min	155.0–1025.0 min
Warm ischemia time (n = 138)	46.0 (49.9) min	46.8–53.1 min	20.0–166.0 min
First Immunosuppression (n = 242)	Tacrolimus n = 203	83.9%	-
	Cyclosporine n = 24	9.9%	-
	mTor inhibitor n = 11	4.5%	-
	Mycophenolate Mofetil n = 1	0.4%	-
	Prednisolone n = 1	0.4%	-
	None n = 2	0.8%	-
Second Immunosuppression (n = 242)	2nd Immunosuppression n = 75	31.0%	
	Mycophenolate Mofetil or Azathioprine n = 44	18.2%	
	Prednisolone n = 27	11.2%	
	mTor inhibitor n = 4	1.6%	
Third Immunosuppression (n = 242)	3rd Immunosuppression n = 16	6.6%	
	Mycophenolate Mofetil or Azathioprine n = 8	3.3%	
	Prednisolone n = 7	2.9%	
	mTor inhibitor n = 1	0.4%	
ANA (n = 241)	negative n = 176	73.0%	-
	1:40 n = 10	4.1%	-
	1:80 n = 15	6.2%	-
	1:160 n = 36	15.0%	-
	≥1:320 n = 4	1.7%	-
SMA (n = 242)	negative n = 116	48.0%	-
	1:20 n = 8	3.3%	-
	1:40 n = 62	25.6%	-
	1:80 n = 47	19.4%	-
	1:160 n = 7	2.9%	-
	≥1:320 n = 2	0.8%	-
LKM (n = 242)	1:20 n = 1	0.4%	-
SLA (n = 241)	positive n = 0	0.0%	-
Any AAB (n = 242)	positive n = 147	60.7%	-
AST (n = 239)	38.0 (57.0) U/L	48.7–65.2 U/L	7.0–670.0 U/L
ALT (n = 242)	31.0 (62.1) U/L	49.2–75.0 U/L	7.0–958.0 U/L
GGT (n = 242)	19.0 (61.4) U/L	48.6–74.2 U/L	6.0–665.0 U/L
INR (n = 242)	1.0 (1.1)	1.0–1.1	0.8–2.1
Total Re-pLTX (n = 242)	n = 17	7.0%	1.0–3.0
Total TCMR (n = 242)	No TCMR n = 135 One TCMR n = 70 Two TCMR n = 27 >2 TCMR n = 10	55.8% 29.0% 11.2% 4.0%	
RAI of liver biopsies with corresponding AAB analysis (n = 112)	2.0 (1.9)	1.60–2.28	0.0–7.0
ISHAK F-latest liver biopsy (n = 91)	1.0 (1.3)	1.0–1.7	0.0–6.0
Interval pLTX-latest liver biopsy (n = 91)	38.8 (57.5) mo	45.9–69.2 mo	0.7–198.8 mo
Study follow-up time (n = 242)	73.5 (82.8) mo	77.3–88.4 mo	15.0–229.0 mo

mo = months, y = years, min = minutes, ULN = upper limit normal, AAB = autoantibody, pLTX = pediatric liver transplantation, TCMR = T-Cell mediated rejection, RAI = rejection activity index, CIT = cold ischemia time.

**Table 2 children-09-00275-t002:** Characteristics associated with SMA titer > 1:20 (odds ratios from binary logistic regression, mutually adjusted for all variables in the model, n = 143).

	Odds Ratio	95% CI for Odds Ratio	*p* Value
Patient sex (male vs. female)	2.30	0.90–5.85	0.081
Patient age at pLTX (per year)	0.94	0.81–1.09	0.428
Donor sex (male vs. female)	0.93	0.38–2.27	0.869
Donor age (per year)	0.97	0.94–0.99	0.046
CIT (per hour)	1.37	1.11–1.69	0.003
pLTX-AABa interval (per month)	0.99	0.98–1.00	0.200
Split liver versus whole organ	2.85	0.60–13.59	0.189
LDLTX versus DDLTX	37.05	5.56–246.77	<0.001
Prednisolone-free immunosuppression	5.28	1.45–19.28	0.012
Aspartate aminotransferase (per U/L)	1.01	0.96–1.03	0.530
Alanine aminotransferase (per U/L)	1.00	0.98–1.01	0.834
Gamma glutamyl transferase (per U/L)	1.00	1.00–1.01	0.200

pLTX = pediatric liver transplantation, CIT= cold ischemia time, AABa = autoantibody analysis, LD = living donor, DD = deceased donor.

**Table 3 children-09-00275-t003:** Characteristics associated with ANA titer > 1:40 (odds ratios from binary logistic regression, mutually adjusted for all variables in the model, n = 142).

	Odds Ratio	95% CI for Odds Ratio	*p* Value
Patient sex (male vs. female)	1.70	0.76–3.80	0.201
Patient age at pLTX (per year)	1.10	0.97–1.25	0.141
Donor sex (male vs. female)	0.73	0.33–1.63	0.447
Donor age (per year)	1.01	0.98–1.05	0.489
CIT (per hour)	1.42	1.18–1.72	<0.001
pLTX-AABa interval (per month)	1.00	0.99–1.01	0.443
Split liver versus whole organ	1.95	0.54–7.05	0.306
LDLTX versus DDLTX	6.37	1.50–27.34	0.013
Prednisolone-free immunosuppression	0.84	0.26–7.70	0.767
Aspartate aminotransferase (per U/L)	1.01	0.99–1.03	0.379
Alanine aminotransferase (per U/L)	1.00	0.98–1.01	0.834
Gamma glutamyl transferase (per U/L)	1.00	0.99–1.00	0.417

pLTX = pediatric liver transplantation, CIT = cold ischemia time, AABa = autoantibody analysis, LD = living donor, DD = deceased donor.

**Table 4 children-09-00275-t004:** Characteristics associated with RAI score (0–9) in multiple linear regression (n = 112).

	Coefficient B (Unstandardized)	95% CI	*p* Value
Patient sex (male vs. female)	0.27	−0.42–0.96	0.445
Patient age at pLTX (per year)	−0.03	−0.12–0.05	0.448
pLTX-AABa interval (per year)	−0.05	−0.14–0.05	0.330
LDLTX vs. DDLTX	−0.48	−1.48–0.52	0.341
SMA titer	0.00	−0.01–0.01	0.466
ANA titer	0.00	0.00–0.01	0.208

pLTX = pediatric liver transplantation, AABa = autoantibody analysis, RAI = rejection activity index, CIT = cold ischemia time.

**Table 5 children-09-00275-t005:** Characteristics associated with RAI score (0–9) in multiple linear regression (n = 112).

	Coefficient B (Unstandardized)	95% CI	*p* Value
Patient sex (male vs. female)	0.27	−0.41–0.95	0.427
Patient age at pLTX (per year)	−0.03	−0.11–0.05	0.475
pLTX-AABa interval (per year)	−0.05	−0.14–0.04	0.292
LDLTX vs. DDLTX	−0.46	−1.44–0.52	0.349
SMA > 1:80 vs. ≤ 1:80	1.12	−0.60–2.85	0.199
ANA > 1:80 vs. ≤ 1:80	0.62	0.29–1.52	0.179

Pltx = pediatric iver transplantation, AABa = autoantibody analysis, RAI = rejection activity index, CIT = cold ischemia time.

**Table 6 children-09-00275-t006:** Characteristics associated with ISHAK fibrosis score (0–6) in multiple linear regression (n = 78).

	Coefficient B (Unstandardized)	95% CI	*p* Value
Patient sex (male vs. female)	0.08	−0.58–0.75	0.802
Patient age at pLTX (per year)	−0.09	−0.20–0.02	0.108
Donor sex (male vs. female)	0.17	−0.50–0.84	0.622
Donor age (per year)	0.04	0.01–0.06	0.009
CIT (per hour)	0.13	0.02–0.23	0.021
pLTX-AABa interval (per year)	0.02	−0.06–0.10	0.664
SMA titer	−0.00	−0.01–0.01	0.569
ANA titer	0.01	0.00–0.01	0.046

pLTX = pediatric liver transplantation, CIT = cold ischemia time, AABa = autoantibody analysis.

**Table 7 children-09-00275-t007:** Characteristics associated with ISHAK fibrosis score (0–6) in multiple linear regression for ANA and SMA Titer >1:80 versus ≤ 1:80 (n = 78).

	Coefficient B (Unstandardized)	95% CI	*p* Value
Patient sex (male vs. female)	0.07	−0.58–0.71	0.838
Patient age at pLTX (per year)	−0.10	−0.20–0.01	0.065
Donor sex (male vs. female)	0.24	−0.42–0.89	0.471
Donor age (per year)	0.03	0.01–0.06	0.009
CIT (per hour)	0.1	−0.01–0.20	0.072
pLTX-AABa interval (per year)	0.02	−0.6–0.10	0.646
SMA >1:80 versus ≤ 1:80	0.05	−1.78–1.89	0.955
ANA >1:80 versus ≤ 1:80	1.10	0.29–1.91	0.009

pLTX = pediatric liver transplantation, CIT = cold ischemia time, AABa = autoantibody analysis.

## Data Availability

Due to the nature of this research, participants of this study did not agree to their data being shared publicly, so supporting data is not available.

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
