# Peer review of "Cold Ischemia Time and Graft Fibrosis Are Associated with Autoantibodies after Pediatric Liver Transplantation: A Retrospective Cohort Study of the European Reference Network TransplantChild"

_children, 2022, doi:10.3390/children9020275_

Round 1

Reviewer 1 Report

This is a brilliant retrospective multicenter study on the development of autoantibodies (AABs) and the correlation thereof with transplant relevant parameters. While in general, the data is clearly presented, the discussion is lacking a critical point by not addressing cost of assessing AABs, the final clinical relevance of AABs and what consequences there are for the treating doctors, in case AABs are being detected. Further, no recommendation is given, whether AABs should be detected routinely and whether this may benefit the patients.

There are two further minor concerns:

  1. Methods, line 73: what was the rationale to use the last transplant after multiple transplants? How can we conclude the AABs are a consequence of the last transplant? Please explain the rationale or exclude these transplants.
  2. Discussion: No discussion of AABs vs. (de novo) DSAs is made, DSAs are not even mentioned and if they were assessed. Similarly, an explanation, why so many patients develop AABs is lacking, both points addressed would give the reader an additional twist.

Author Response

We thank the reviewers extraordinarily for their detailed and very helpful comments. We could improve the manuscript substantially based on their recommendation.

This is a brilliant retrospective multicenter study on the development of autoantibodies (AABs) and the correlation there of with transplant relevant parameters. While in general, the data is clearly presented, the discussion is lacking a critical point by not addressing cost of assessing AABs, the final clinical relevance of AABs and what consequences there are for the treating doctors, in case AABs are being detected. Further, no recommendation is given, whether AABs should be detected routinely and whether this may benefit the patients.

We agree that clinical relevance and impact of AAB needs further examinations. However, our study provides initial evidence that AB could be seen as marker proteins indicative of a latent graft injury/ silent graft fibrosis. Whilst the mechanism and the specificity is not clear so far they seem to be a good screening parameter, inexpensive to measure (around 10 Euro for each) and may be helpful to modify immunosuppression. That means our recommendation is to measure them once a year or by clinical indication (elevated liver enzymes). This reflects our standard of care at our center. We suggest to be alert if antibodies (ANA or SMA) are present (at least high titer >1:80). Especially if reduction of immunosuppression is planned, indication for liver biopsy should be discussed. However, these points need to be handled with caution, further studies would be necessary to confirm these results, ideally with standard follow up biopsies. If studies with follow up biopsies are done we strongly recommend analyzes of AAB simultaneously. We revised the manuscript according to the reviewer comment and added these aspects.

There are two further minor concerns:

  1. Methods, line 73: what was the rationale to use the last transplant after multiple transplants? How can we conclude the AABs are a consequence of the last transplant? Please explain the rationale or exclude these transplants.

Including earlier transplant and evaluating outcome later -after a second transplant- would make it impossible to evaluate the association of donor and transplant characteristics with outcome, since donor and characteristics from first LT would be included and outcome would be measured after last transplant (depending om time in between). We cannot exclude that AABs are from first LT, but we can also not exclude AAB positivity before LT. However, the aim of our study was to evaluate the presence of AAB and potentially association, independently of AAB presence before LT. To evaluate also AAB before transplant in a multicenter prospective study would be great but is only a realistic approach, if existing data indicating a meaning of AABs are available. That is what our study was made for. We added these thoughts to the manuscript.

  1. Discussion: No discussion of AABs vs. (de novo) DSAs is made, DSAs are not even mentioned and if they were assessed. Similarly, an explanation, why so many patients develop AABs is lacking, both points addressed would give the reader an additional twist.

DSAs were out of the scope of this study and were not assessed. We agree this is an interesting aspect and worth exploring in future studies. We could only speculate if DSA would be expressed in a similar fashion. We added a further sentence to the part, where we discuss the generation of AAB.

Reviewer 2 Report

The authors have provided a multi-center study, with a nicely described methodology and meticulously driven statistical analysis. However, the manuscript requires major revision, in accordance with the following suggestions:

  1. Abstract: the aim of the study is not clearly stated ("We aimed to study prevalence, association with donor-, transplant-, recipient-charac- 33
    teristics and relation with outcome in a multicenter study to clarify the contradictions."). Which contradictions are you talking about?
  2. "Independently of interval between pLTX and AAB analysis risk factors 36
    for SMA and ANA positivity were cold ischemia time (CIT) with Odds Ratio (OR) 1.37 (95% CI 1.11- 1.69) and 1.42 (95%CI 1.18-1.72), respectively per one hour increase in CIT and for SMA positivity steroid-free immunosuppression (IS) vs. IS including steroid medication (OR 5.28; 95% CI 1.45- 19.28)." This phrase from the abstract which tries to briefly depict main results is too long and hard to follow. Please rephrase it and divide it into shorter sentences.
  3. Line 58: "other diagnostic aspects". Which other diagnostic aspects?
  4. Lines 70-71: "6 European transplant centers within the European Reference Network (ERN) TransplantChild". Please name this 6 centers
  5. Line 85:I think the authors should detail what rejection activity index (RAI) and ISHAK Fibrosis score are composed of.
  6. Discussion section: Reference to tables should be provided only among the results chapter.
  7. As a limitation the retrospective design of the study is worth discussing.
  8. Exclusion criteria were not stated.
  9. Introduction is too short. Present knowledge on the subject could be detailed and enriched with other references as well.
  10. I personally believe that the construction "but not cellular rejection" could be let out of the title.
  11. English language requires corrections.
  12. An institutional review board approval has not been provided. If not needed, please state this fact.
  13.  

Author Response

We thank the reviewers extraordinarily for their detailed and very helpful comments. We could improve the manuscript substantially based on their recommendation.

The authors have provided a multi-center study, with a nicely described methodology and meticulously driven statistical analysis. However, the manuscript requires major revision, in accordance with the following suggestions:

  1. Abstract: the aim of the study is not clearly stated ("We aimed to study prevalence, association with donor-, transplant-, recipient-charac- 33
    teristics and relation with outcome in a multicenter study to clarify the contradictions."). Which contradictions are you talking about?

We were talking about the conflicting results from earlier studies. Some of them showed an association of AAB with outcome and some not. We revised this sentence in the abstract.

  1. "Independently of interval between pLTX and AAB analysis risk factors 36
    for SMA and ANA positivity were cold ischemia time (CIT) with Odds Ratio (OR) 1.37 (95% CI 1.11- 1.69) and 1.42 (95%CI 1.18-1.72), respectively per one hour increase in CIT and for SMA positivity steroid-free immunosuppression (IS) vs. IS including steroid medication (OR 5.28; 95% CI 1.45- 19.28)." This phrase from the abstract which tries to briefly depict main results is too long and hard to follow. Please rephrase it and divide it into shorter sentences.

We revised the sentence according to the recommendation of the reviewer.

Independently of interval between pLTX and AAB analysis one hour increase in CIT resulted in a Odds ratio (OR) of 1.37 (95% CI 1.11-1.69) for SMA positivity and in OR of 1.42 (95%CI 1.18-1.72) for ANA positivity. Steroid-free immunosuppression (IS) vs. steroid-including IS (OR 5.28; 95% CI 1.45-19.28) was a risk factor for SMA positivity

  1. Line 58: "other diagnostic aspects". Which other diagnostic aspects?

With this statement, we intended to refer to the Autoimmune Hepatitis Score.

We revised the sentence, to make it more precise.

  1. Lines 70-71: "6 European transplant centers within the European Reference Network (ERN) TransplantChild". Please name this 6 centers

We added the names. (Hannover, Paris, Bergamo, Warsaw, Vilnius, Rome)

  1. Line 85:I think the authors should detail what rejection activity index (RAI) and ISHAK Fibrosis score are composed of.

We added this information the supplement and referred to the supplement in the manuscript.

  1. Discussion section: Reference to tables should be provided only among the results chapter.

We deleted the reference to the tables in the discussion section.

  1. As a limitation the retrospective design of the study is worth discussing.

We agree with the reviewer. Our sentence about the variable AAB analysis-pLTX time intended to include the limitation of retrospective design. We revised this section to make it clearer.

  1. Exclusion criteria were not stated.

We completely agree with the reviewer. Sorry that we forgot this important point. We added to the manuscript foolowing sentence: “Patients who died within the first three months or only have AABa within the first three months after pLTX and patients with ANA or SMA analysis by ELISA only were excluded. Patients with combined transplantations were excluded.”

  1. Introduction is too short. Present knowledge on the subject could be detailed and enriched with other references as well.

We revised the introduction and added some details/aspect and also some more references.

  1. I personally believe that the construction "but not cellular rejection" could be let out of the title.

In the beginning we felt that this important information should the readers be given in the title. Now we agree with the reviewer, that the title is more clear and comprehensible without this construction. We changed title to:  “Cold ischemia time and graft fibrosis are associated with auto-antibodies after pediatric liver transplantation: a retrospective cohort study of the European Reference Network TransplantChild”

  1. English language requires corrections.

We revised the manuscript/language based on recommendations of a native speaker.

  1. An institutional review board approval has not been provided. If not needed, please state this fact.

We added following sentences to the method section: “This observational study was performed completely anonymous and retrospectively, therefore ethical approval was not necessary. “

Round 2

Reviewer 2 Report

The authors did a great job in following my recommendations.

This manuscript is a resubmission of an earlier submission. The following is a list of the peer review reports and author responses from that submission.

Round 1

Reviewer 1 Report

This is a large multi-centre dataset and well analysed. Data is presented clearly. The data reinforces current understanding that denovo AAB post LTx are present and are related in someway to graft fibrosis. The correlation with prolonged CIT is a clear finding in your data. What are your thoughts about the cause of these findings? 

Author Response

Thanks to the reviewer for reviewing this manuscript and the helpful comments. The correlation of CIT with AAB is observation in a retrospective study of a mixed cohort. It is not possible to point this correlation on one cause. Based on our study we are not able to prove any causative aspects. We can only describe the observation and that this observation/ association is stronger than all other effects influencing prevalence of AAB.  Nevertheless, we added to the discussion some sentence about possible pathophysiological mechanism.

Reviewer 2 Report

This is a well written manuscript and the results have clinical implications but there are several concerns.

  1. The authors suggested that the prevalence and the titer of AAB were strongly associated with CIT. However, the prevalence and the titer of AAB varies over time after the transplantation and this could affect the association between AAB and CIT. And in Center differences part, description of age at pLTX and AAB analysis in each center would be helpful. 
  2. The authors suggested that there was a strong association between AAB and CIT. Would you explain the possible mechanism of the association? 

Author Response

Thanks to the reviewer for reviewing this manuscript and the helpful comments.

Reply to Point 1.): The reviewer is right the prevalence and titer of AAB varies over time and increase with longer time from pLTX and time of AAB analyses in our study was not standardized since it is a retrospective study, nonetheless the fact that the association is still significant, even in this mixed cohort, makes this result even stronger in our opinion.  We added data for age at pLTX, age at AABa and interval between both for each center to the table 5. We also describe the limitation of the not standardized time point in our limitation paragraph.

Reply to Point 2.):We added to the discussion some sentences in which we explain/discuss possible mechanism.

Reviewer 3 Report

The autoantibody levels were examined in the pediatric liver transplantation recipients and the authors found that they were associated with the cold ischemic time and graft fibrosis.

I have some comments.

  1. (P2, L77) It is described that “One AAB measurement at any time after pLTX (by indication or by protocol) was included for every patient.” How about the data on the between the measurement and the operation?
  2. (Discussion section) Please add a paragraph on the limitation of the present study.

Author Response

Thanks to the reviewer for reviewing this manuscript and the helpful comments.

Reply to Point 1.): This study aimed to evaluate the prevalence of AAB in a large pLTX cohort on to test for possible risk factors and associated outcome parameter. Since it is a retrospective study, we have been dependent on available data. This is the reason why the time of AAB analysis was not standardized and the reason why we do not have longitudinal measurements for the cohort. AAB analysis was not routinely done in many patients. But for the aim of our study it was sufficient to know, if AAB are positive in a patient at any time. Furthermore, we evaluated a subgroup were AAB analysis was timely connected to liver biopsy, which was important to test for association of AAB and T cell mediated rejection. We also describe the limitation of the not standardized time point in our limitation paragraph.

Reply to Point 2.): Obviously, we did not point out enough our limitation paragraph. We revised our limitation part and tried to make it more visible.

Round 2

Reviewer 2 Report

The revised manuscript is improved and the authors have addressed the comments of the reviewer. 

Author Response

Thanks to the reviewer for reviewing the manuscript again.

Reviewer 3 Report

The autoantibody levels were examined in the pediatric liver transplantation recipients and the authors found that they were associated with the cold ischemic time and graft fibrosis.

I have a comment.

1 What is the relationship between the duration between LTx and AAB analysis and AAB positivity?

Author Response

Thanks to the reviewer to review the mansucript again. Sorry, but we are not sure about the question of the reviewer. If we understand right, the reviewer wants to know, if AAB positivity is associated with interval between LTX und AAB measurement, right? This we have described in our manuscript at paragraph 3.4.3 and in table 3.

Round 3

Reviewer 3 Report

The autoantibody levels were examined in the pediatric liver transplantation recipients and the authors found that they were associated with the cold ischemic time and graft fibrosis.

I think that the duration between LTx and AAB analysis should be prospectively set. If it was short and the AAB was negative in one patient, the results can turn positive in the longer period when the AAB of the other patients were analyzed. In such a situation the multivariate analysis with the AAB results as a x factor seems to have little significance.